# Monocyte-cancer cell fusion is mediated by phosphatidylserine—CD36 receptor interaction and induced by ionizing radiation

**Ivan Shabo**[1,2]*, **Kristine Midtbö**[3], **Robert Bränström**[1,2], **Annelie Lindström**[3]

**1** Endocrine and Sarcoma Surgery Unit, Department of Molecular Medicine and Surgery, Karolinska Institutet, Stockholm, Sweden, **2** Department of Breast Cancer, Sarcoma and Endocrine Tumors, Theme Cancer, Karolinska University Hospital, Stockholm, Sweden, **3** Division of Cell- and Neurobiology, Department of Biomedical and Clinical Sciences, Faculty of Medicine and Health Sciences, Linköping University, Linköping, Sweden

* ivan.shabo@ki.se

**Data Availability Statement:** All relevant data are within the paper and its Supporting Information files.

## Abstract

Emerging evidence suggests that fusion of cancer cells with leucocytes, such as macrophages, plays a significant role in cancer metastasis and results in tumor hybrid cells that acquire resistance to chemo- and radiation therapy. However, the precise mechanisms behind the leukocyte-cancer cell fusion remain unclear. The present *in vitro* study explores the presence of fusion between the monocyte cell line (THP-1) and the breast cancer cell line (MCF-7) in relation to the expression of CD36 and phosphatidylserine with and without treatment of these cells with ionizing radiation. The study reveals that spontaneous THP-1/MCF-7 cell fusion increases significantly from 2.8% to 6% after irradiation. The interaction between CD36 and phosphatidylserine plays a pivotal role in THP-1/MCF-7 cell fusion, as inhibiting this interaction using anti-CD36 antibodies significantly reduces cell fusion. While irradiation leads to a dose-dependent escalation in phosphatidylserine expression in MCF-7 cells, it does not impact the expression of CD36 in either THP-1 or MCF-7 cells. To the best of our knowledge, this is the first study to demonstrate the involvement of the CD36-phosphatidylserine interaction in the fusion between monocytes and cancer cells, shedding light on a novel explanatory mechanism for the roles of CD36 and phosphatidylserine in tumor progression.

## Introduction

Cell fusion is a fundamental evolutionarily conserved biological process in mammalian tissue development and differentiation during embryogenesis and morphogenesis [1]. Cell fusion also plays a significant role in tissue repair and regeneration, e.g., liver, heart, and intestine [2–4]. This process yields viable pluripotent hybrid cells that acquire nuclear reprogramming and epigenetic modifications with new genetic and phenotypic properties at a rate exceeding that is achievable by random mutations [4–6].

**Funding:** "The Research Council of Southeast Sweden Project ID: FORSS-229761" were used to pay for research materials in the present study. All authors agree that study data can be shared upon acceptance for publication.

**Competing interests:** No. The authors have no competing interests.

Macrophages are a heterogeneous population of cells derived from monocytes [7] and exhibit two different polarisation states, M1 and M2, in response to various signals in the tissue microenvironment. M1-macrophages are pro-inflammatory and characterized by the release of inflammatory cytokines. M2-macrophages are immunosuppressive, have a scavenging function, and support tissue repair. In cancer, macrophages promote tumor cell migration and metastasis. Tumor-associated macrophages represent the M2-macrophage and promote tumor progression [8–12].

Fusion is a crucial function of macrophages, playing an essential role in forming, *e.g.*, osteoclasts and multinucleated giant cells. Macrophage/cancer cell fusion results in cellular reprogramming and generates new tumor cell subsets that acquire the ability for chemotactic migration by combining the epigenetic program of the leucocyte with the uncontrolled cell division of the cancer cell [6, 13, 14] (Fig 1). Tumor cells are also fusogenic. Spontaneous fusion among cancer cells is a well-documented phenomenon in solid tumors and generates heterogeneous subpopulations of tumor cells [15–18]. During the last 15 years, comprehensive *in vitro* [19, 20], *in vivo* [21, 22], and clinical evidence [23–27] shows that the fusion of cancer cells with leucocytes, such as macrophages, plays a significant role in cancer metastasis and results in tumor hybrid cells that acquire resistance to chemo- and radiation therapy.

CD36 is a scavenger receptor physiologically expressed by multiple cells, including adipocytes, macrophages, and endothelial cells [28]. It selectively binds ligands to proteins such as

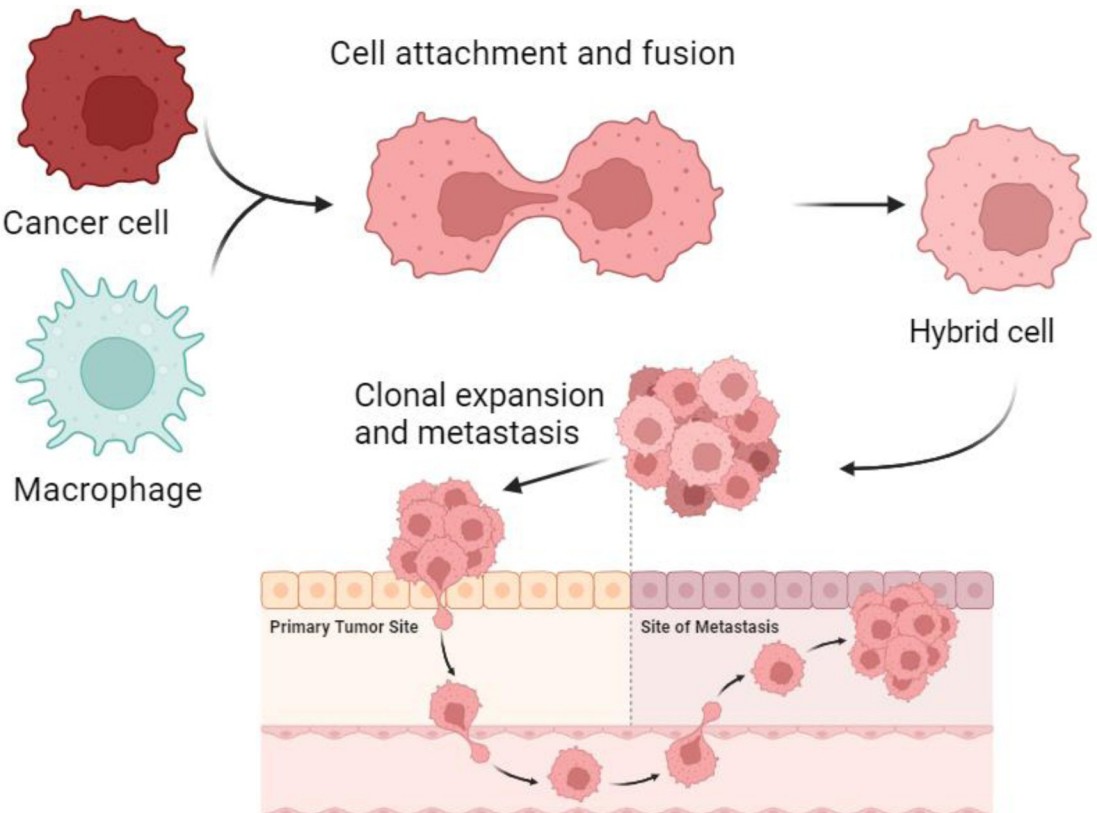

**Fig 1. A schematic diagram of the cell fusion process, hybrid formation and metastasis.** Based on cell fusion theory, a macrophage is drawn to a cancer cell. The outer cell membranes of the two cells become attached. Fusion occurs by initially forming a bi-nucleated heterokaryon with a nucleus from each of the fusion partners and later goes through genomic hybridization, creating a macrophage/cancer cell-derived hybrid cell with genetic and phenotypic traits from both maternal cells.

oxidized phospholipids and thrombospondins [29–31]. Several cellular stimuli, such as lipo-proteins, reactive oxygen species, and inflammation, regulate the expression of CD36 in macrophages. As a scavenger and recognition receptor, CD36 is involved in the clearance of cell debris and phagocytosis. CD36 expression is also reported in different breast cancer cell lines, such as BT-483, HCC2218, MCF-7, and MDA-MB-468, and it is associated with enhancing their proliferative and invasive activities [32–34].

Phosphatidylserine (PS) is a glycerophospholipid, normally located in the cytosolic leaflet of the cell membrane. However, when a cell undergoes apoptosis, PS relocates to the extracellular surface of the cell membrane, serving as a signal for macrophages to engulf the apoptotic cells. During phagocytosis, the macrophage uses CD36 to detect PS expressed by apoptotic cells. Recently, the interaction between PS and CD36 has been reported to be a crucial step in the fusion between monocytes and tumor cells [35–37].

This study explores the impact of ionizing radiation on CD36 and PS expression in cancer cells and monocytes. Furthermore, we investigate whether blocking the interaction between CD36 and PS will prevent fusion between monocytes and cancer cells.

## Material and methods

### Cell culture

THP-1 and MCF-7 (ATCC, Manassas, USA) cells were cultivated in complete RPMI medium (+25 mM HEPES, + Glutamax) supplemented with 10% fetal bovine serum (FBS) (Gibco Life Technologies, New York, USA) and 1% penicillin-streptomycin (PEST) (Thermo Scientific, Waltham, USA), at 37˚C with 5% $CO_2$. The cells were seeded in T75 tissue culture flasks (Sigma-Aldrich Co, ST. Louis, USA) and passaged 1:3 twice a week. Cell viability was estimated using trypan blue (Thermo Scientific, Waltham, USA).

### CellTrace™ staining

THP-1 and MCF-7 cells were washed in PBS (Jena Bioscience, Jena, Germany), and the cell density was adjusted to $1x10^6$ cells/ml. The THP-1 cells were stained with the CellTrace™ tracer dye Violet (V) (excitation/emission maxima: 405/450, Life Technologies, C34557, Carlsbad, USA) and MCF-7 cells with Far Red (FR) (excitation/emission maxima: 630/661, Life Technologies, C34564, Carlsbad, USA) for 30 minutes in darkness at 37˚C with 5% $CO_2$. The concentrations of dyes were optimized in relation to two days of cultivation. The cells were stained with 0.1–10 μM V and 0.01–5 μM FR, respectively, and incubated with a complete medium, five times the staining volume, for 5 minutes in darkness at 37˚C (Fig 2). Followed by a three-time washing procedure in complete medium, the cells were centrifuged at 320 g for 5 minutes at room temperature.

### Cell fusion

To induce spontaneous cell fusion, the THP-1 cell labelled with CellTrace™ tracer dye Violet and MCF-7 cells labelled with CellTrace™ tracer dye Far Red were co-cultured in the same cell culture vial in RPMI 1640 medium for two days. The cells were seeded at a ratio of about 1:3 THP-1 to MCF-7. The harvested cells were analyzed using BD FACSAria™ III (BD Bioscience, USA) and compared to unstained cells (negative control) as well as stained cells grown in single cultures (positive controls). Cells positive for both FR and V, THP1/MCF-7 tumor hybrid cells, were sorted, collected, and cultivated in a complete RPMI medium supplemented with 20% FBS. Cell fusion experiments were repeated nine times (Fig 3). All experiments were repeated three times.

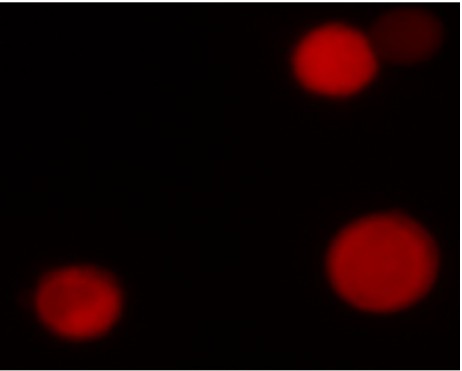

**A**

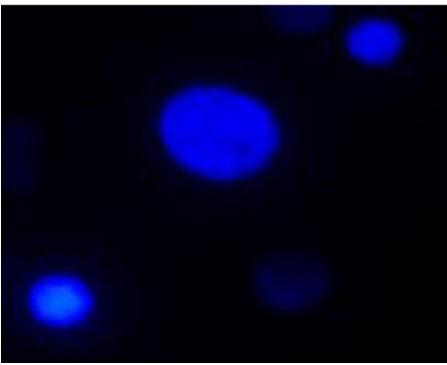

**B**

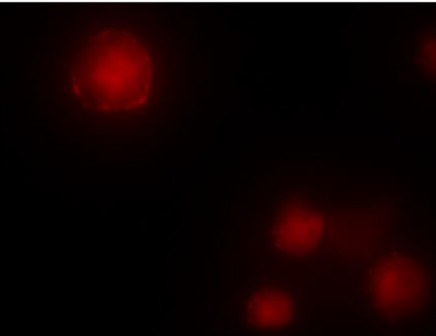

**C**

**Fig 2.** Fluorescent microscopy of CellTrace and CD36 staining A) MCF-7 cells stained with CellTrace Far red B) THP-1 cells stained with CellTrace Violet C) THP-1 cells stained with CD36 antibodies. All images were taken with x40 magnification.

## Prevention of cell fusion with anti-CD36 antibodies

Anti-CD36 antibody (mouse monoclonal IgG1 from Life Span BioScience, LS-C134826, Seattle, USA), 40 μg and 80 μg per ml complete RPMI medium, was used to block CD36. The

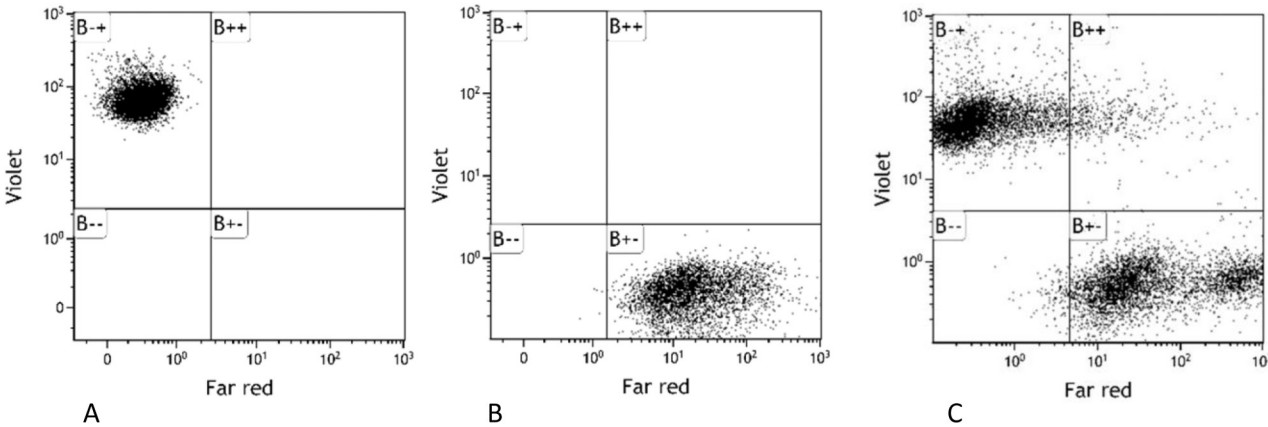

**Fig 3. Flow cytometry analysis of cells stained with CellTrace™ tracer.** (A) THP-1 cells stained with CellTrace Violet, (B) MCF-7 cells stained with CellTrace Far Red, and (C) Co-culture of Stained THP-1 and MCF-7 cells where double positive THP-1/MCF-7 tumor hybrid cells seen in gate B++.

antibody was added to newly started co-cultures of THP-1 and MCF-7 cells at 37°C and 5% $CO_2$. The co-cultures were supplemented with aliquots of antibody solution twice daily for two days. The percentage of double-positive cells was identified using FACS ARIA III and compared to a control co-culture grown without a supplement of anti-CD36 antibody. The experiment was performed with triplicate samples.

## Treatment with gamma irradiation

To explore the effect of ionizing radiation on CD36 and PS expression in THP-1 and MCF-7 cells individually, the cells were cultured in separate cell culture flasks and exposed to increasing doses of 0 (control), 2.5, 5, and 10 Grey (Gy). Low-LET γ-irradiation was administrated via Clinac 600C/D (Varian Medical Systems, Palo Alto, USA) as a radiation source with one AP field, using linear accelerated 6 MV photons, dose rate 5 Gy/minute. The cells were surrounded by 3 cm Poly methyl methacrylate build-up material and ice. The cells were seeded into T75 flasks one day before the irradiation procedure at cell densities of $1.2 \times 10^6$ MCF-7 cells and $1.5 \times 10^6$ THP-1 cells per flask. The cells were harvested for flow cytometry (BD FAC-SAria™ III) analysis at 1 hour and 4 hours after irradiation (Fig 4).

To study the impact of ionizing radiation on cell fusion events, the THP-1 cell labeled with CellTrace™ tracer dye Violet and MCF-7 cells labeled with CellTrace™ tracer dye Far Red were co-cultured in the same cell culture vial in RPMI 1640 medium and treated with 0 Gy (control) and 5 Gy low-LET γ-irradiation, respectively. The cells were cultured for two days and analyzed for detection of cell fusion events according to the previously described cell fusion protocol (Fig 4).

Cell viability rates for MCF-7, THP-1, and tumor hybrid cells in experiments without irradiation were 94%, 59%, and 84%, respectively. After irradiation, the corresponding rates for MCF-7 and THP1 cells were 94% and 45%, respectively. The rate of tumor hybrid cells generated in experiments with irradiation was 68%. The experiment was performed with triplicate samples.

## CD36 and PS expression analysis

CD36 expression was analyzed in MCF-7 and THP-1 monocytes for irradiated cultures and corresponding controls *(0 Gy)*. The cell cultures were washed in PBS, incubated with Human TruStain FcX™ (Fc Receptor Blocking Solution) (BioLegend, 422302, San Diego, USA) for 10

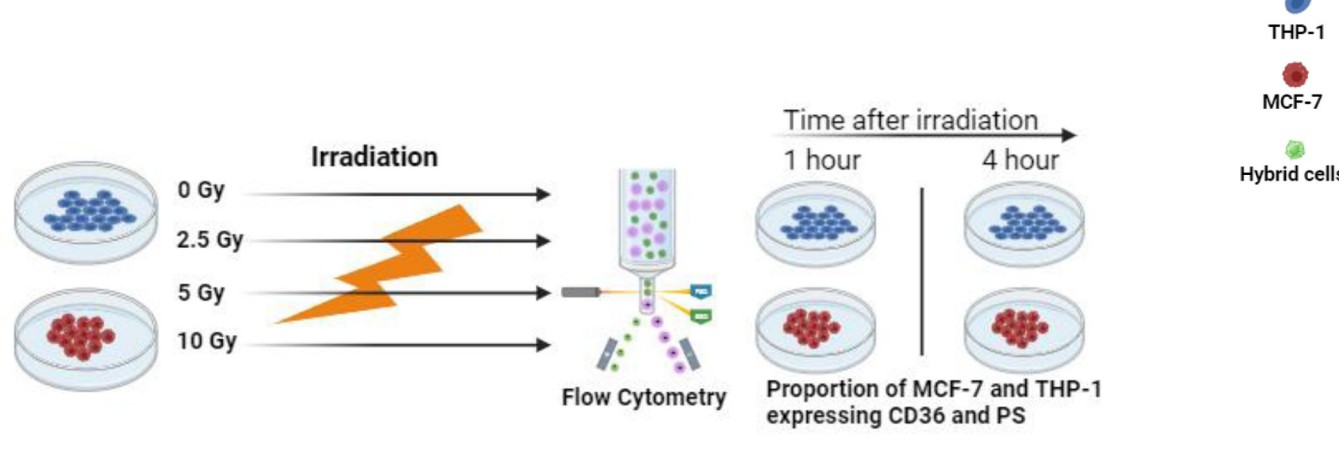

**A**

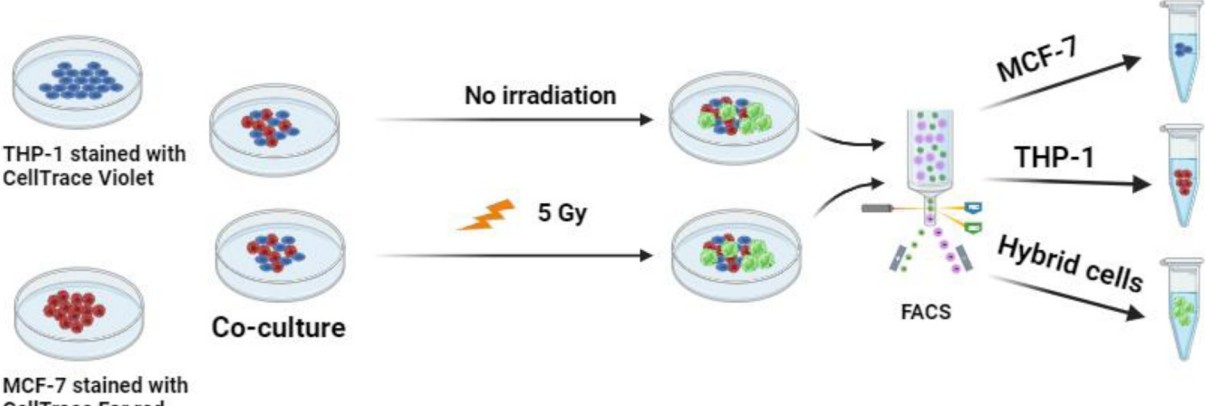

**B**

**Fig 4. Experimental model of THP-1/MCF-7 cell fusion in relation to phosphatidylserine (PS) and CD36 expression in both cell types, with and without gamma irradiation.** Panel A shows THP-1 and MCF-7 cells cultured in separate flasks and treated with 0, 2.5, 5, and 10 Gy radiation. After that, the cells were analyzed by flow cytometry to detect the proportion of cells expressing PS and CD36. Panel B shows a cell fusion experiment where THP-1 cells (labelled with CellTrace Violet) and MCF-7 cells (labelled with CellTracer Far Red) are grown in the same flask and treated with 0 and 5 Gy gamma radiation. After 2 days of co-culture, the cells are sorted with FACS Aria III. Cells co-expressing CellTrace Violet and Far Red are assessed as THP1-/MCF-5 tumor hybrid cells.

minutes at 4˚C, 1:20 ratio to Cell Staining Buffer (Nordic Biosite, ASB-022501L, Täby, Sweden), followed by staining with 5 μg/ml Anti-Human CD36/SCARB3 Antibody (Phycoerythrin (PE)-conjugated) (Rabbit monoclonal IgG antibody, Sino Biological, 10752-R001-P,

Beijing, China), or 50 µg/ml Rabbit IgG Isotype Control, PE-conjugated (Bioss, bs-0295P-PE, Massachusetts, USA) for 30 minutes in darkness at 4˚C. The samples were washed three times in Cell Staining buffer before resuspension in PBS and analyzed according to FACS ARIA III.

For PS expression analysis, the irradiated and control cell cultures were washed in PBS, and the cells were stained with ApoDETECT[TM] Annexin V-FITC kit (Invitrogen[TM], Life Technologies, 33–1200, Stockholm, Sweden) in a 1:20 ratio between Annexin V-FITC and 1X Annexin Binding Buffer, for 10 minutes in darkness at room temperature. The cells were washed once, resuspended in Annexin Binding Buffer, and analyzed according to FACS ARIA III.

### Immunofluorescence microscopy

The MCF-7 and THP-1 cells were seeded on coverslips and incubated for 24 h in RPMI + 10% FBS. For staining with CD36 antibody, THP-1 cells were fixed with 4% paraformaldehyde for 30 min at 37˚C, washed once in PBS followed by permeabilization/blocking for 30 min in 2% BSA/0.1% Saponin in PBS. The THP-1 cells were then incubated with a mouse monoclonal CD36 antibody in PBS/0.5% BSA for 2 h at room temperature and washed three times with PBS. A secondary antibody goat anti-mouse IgG Alexa Fluor 546 (Invitrogen) was added in PBS/0.5% BSA for 45 min, followed by three washes with PBS. After staining of THP-1 cells with the CellTraceTM tracer dye Violet (V) and MCF-7 cells with Far Red (FR) for 30 minutes in darkness. THP-1 and MCF-7 cells were washed three times with PBS. The cover slips were mounted on microscope slides in Dako fluorescence mount media and images were taken with a Zeiss Axiovert 200 M fluorescence microscope with a Zeiss Plan-APOCHROMAT 63x/1.4 oil DIC objective. The images were taken with x40 magnification.

### Statistical analysis

Statistical analyses were performed using SPSS statistics software, version 28 (IBM Corporation, USA). Non-parametric Mann-Whitney U-test was used to test the statistical significance of differences between means of expression CD36 and PS among THP-1 and MCF-7 cells as well as the proportions of cell fusion events in relation to irradiation doses (0, 2.5, 5, and 10 Gy) and recovery time, 1 and 4 hours respectively. A P-value < 5% was considered as statistically significant.

## Results

### Radiation induces the exposure of PS on MCF-7 but not on THP-1 cells

The expression of PS in THP-1 cells was not dependent on radiation dose nor post-irradiation time of 1 and 4 hours (P = 0.9). Analyzing the cells 1 hour after irradiation, PS was expressed in 3.9% of non-irradiated cells. The proportions of PS-positive THP-1 cells after irradiation with 2.5, 5, and 10 Gy were 6.5%, 7.2% and 6.3%, respectively. The corresponding rates of PS expressing THP-1 cells analyzed 4 hours after irradiation were 9% in non-irradiated cells and 9,75%, 7,7%, and 9,9% of irradiated cells with 2.5, 5, and 10 Gy, respectively (Fig 5).

For MCF-7 cells, the proportion of PS-positive cells was radiation dose-dependent and increased consistently to radiation doses of 2.5, 5, and 10 Gy (P = 0.048). The expression of PS was higher in MCF-7 cells analyzed 4 hours after irradiation compared to those diagnosed after 1 hour. The mean proportion of PS-positive cells in non-irradiated MCF-7 cultures was 6.4% (SD 0.014). The highest detected rate of PS-positive MCF-7 cells was 26.2% in the samples irradiated with 10 Gy and analyzed 4 hours after exposure to the radiation (Fig 5).

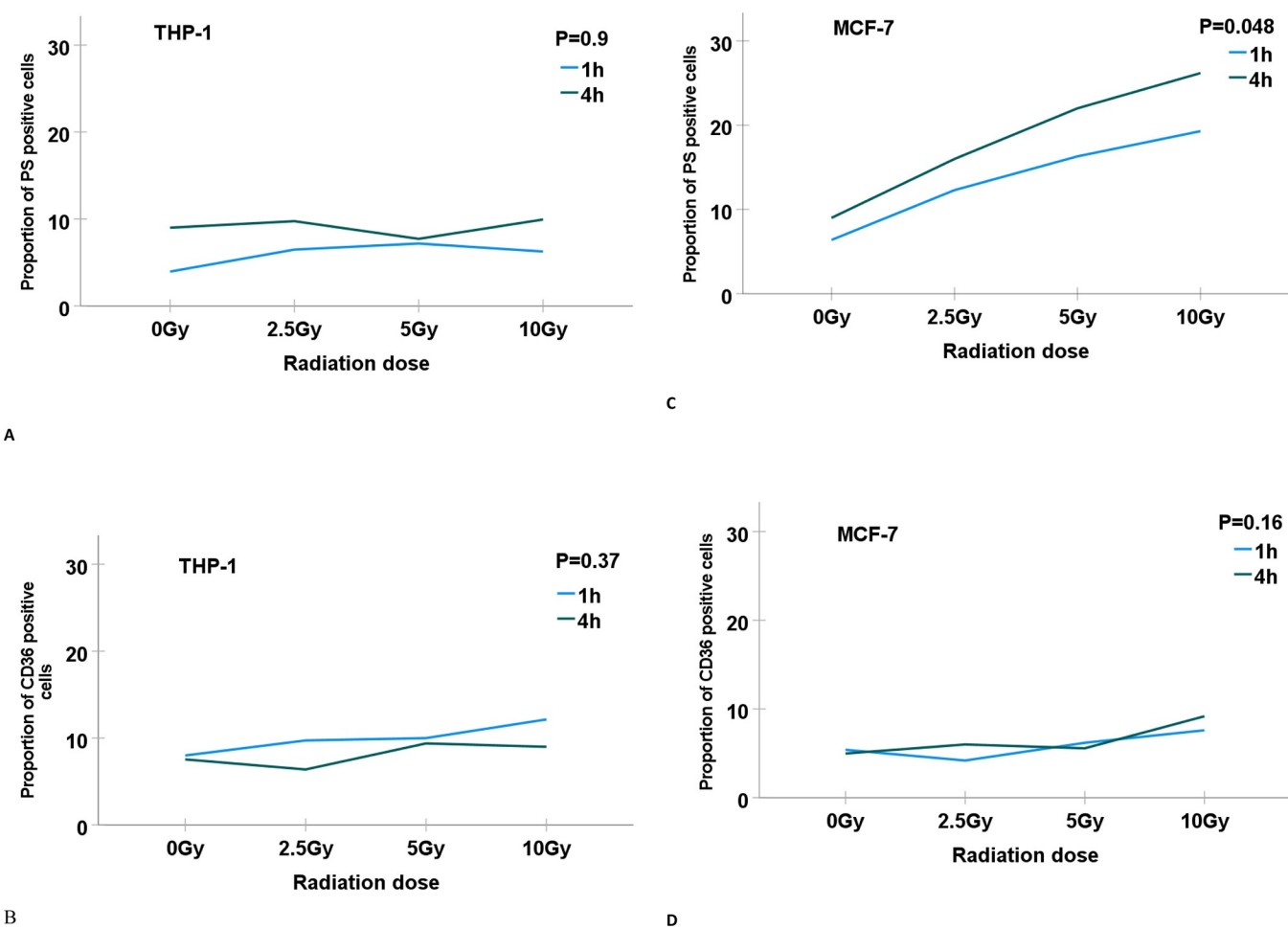

**Fig 5. The proportion of THP-1 and MCF-7 cells expressing phosphatidylserine (PS) and CD36 after treatment with gamma radiation (0, 2.5, 5, and 10 Gy).** The proportion of MCF-7 expressing PS increases in dose-dependent relation (C). No correlation of PS and CD36 expression in THP-1 cells nor CD36 expression in MCF-7 cells was found in relation to the dose of ionizing radiation.

## CD36 expression in MCF-7 and THP-1 cells

One hour after the irradiation procedure, CD36 was expressed in 5.4% of non-irradiated MCF-7 cells and 4.2%, 6.2%, and 7.6% of irradiated cells with 2.5, 5, and 10 Gy, respectively. The corresponding rates of CD36 expression analyzed 4 hours after irradiation were 5% for non-irradiated cells and 6%, 5.6%, and 9% for irradiated MCF-7 cells, respectively. These differences in the proportion of MCF-7 expressing CD36 in relation to different radiation doses and time after irradiation were not statistically significant (P = 0.16) (Fig 5).

For THP-1 cells, the proportions of cells expressing CD36 were 8% of non-irradiated cells and 9.7%, 10%, and 12% of irradiated cells with 2.5, 5, and 10 Gy, at 1 hour after irradiation. The proportions of THP-1 cells expressing CD36 analyzed 4 hours after irradiation were 7.6% of non-irradiated cells, 6.4%, 9.4%, and 9% of cells of irradiated cells, respectively (P = 0.37) (Fig 5).

## THP-1/MCF-7 fusion events in relation to irradiation

After two days of co-culturing THP-1 and MCF-7 cells, 2.8% of cells expressed both CellTraceTM tracer dyes V and FR and were defined as THP-1/MCF-7 tumor hybrid cells. After

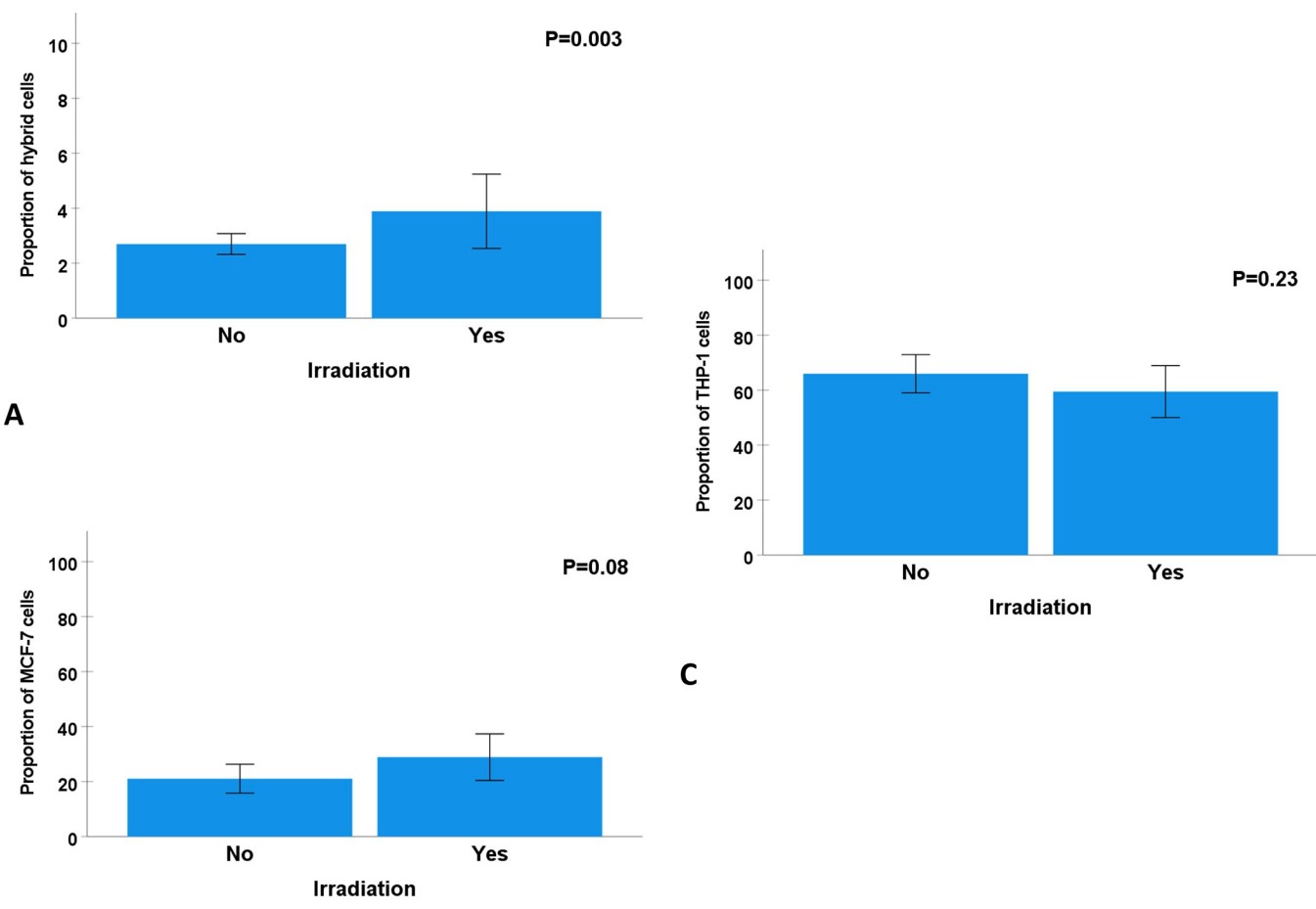

**Fig 6.** THP-1/MCF-7 cell fusion events (A), proportion of MCF-7 (B) and THP-1 cells (C) in relation to gamma radiation (RT). The fusion rate was significantly higher in irradiated (6%) THP-1 and MCF-7 co-culture compared to non-irradiated cells (2.85%). No differences in the proportion of MCF-7 cells (B) nor THP-1 cells were found in relation to irradiation.

treating the THP-1 and MCF-7 cells co-cultures with 5 Gy irradiation, the proportion of tumor hybrid cells increased significantly to 6% (P = 0.003) (Fig 5). Two days after co-culture, there was no significant difference in the proportions of THP-1 and MCF-7 cells in the co-cultures with irradiation compared to non-irradiated cell cultures (Fig 6).

## Cell fusion is inhibited by blocking of CD36-PS interaction

The proportion of tumor hybrid cells in co-cultures supplemented with 40 μg/ml and 80 μg/ml CD36 antibody was 1.8% and 1.9%, respectively. In non-treated, i.e., without CD36 antibody, THP-1/MCF-7 co-cultures, the proportion of hybrid cells was 3.6%, which is significantly higher compared to co-cultures treated with anti-CD36 antibody (P<0.001) (Fig 7).

## Discussion

The fusion between cancer cells and leukocytes, such as monocytes/macrophages, has been suggested as an alternative pathway for cancer cells to acquire exogenous genetic material and develop new phenotypic and functional traits [38]. Currently, the research on cell fusion primarily focuses on detecting hybrid cells in solid tumors and their clinical significance and

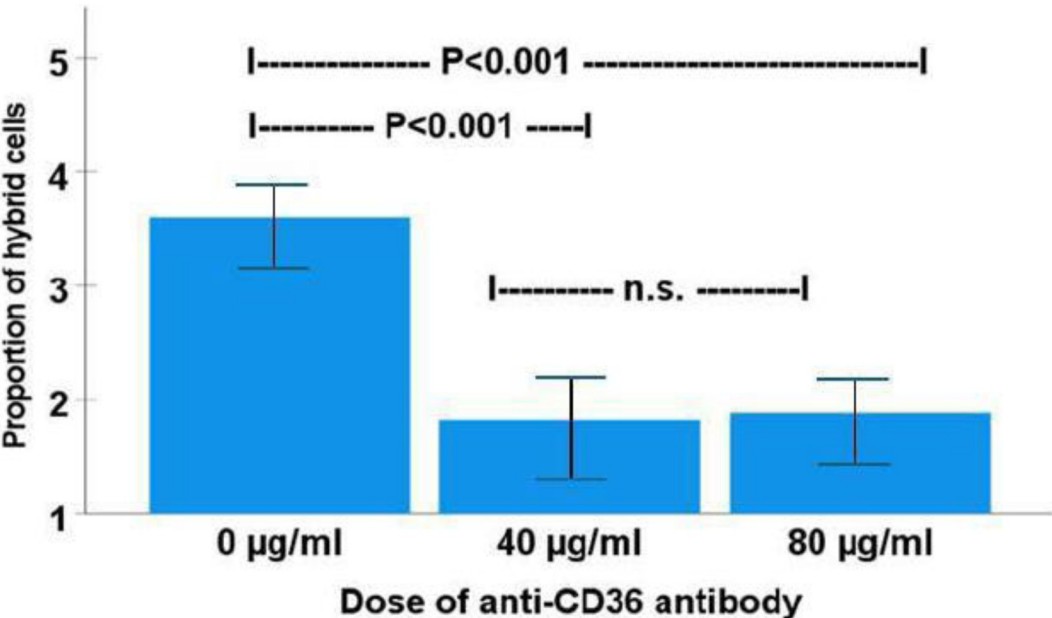

**Fig 7. THP-1/MCF-7 cell fusion rate in relation to treatment with anti-CD36 antibody.** The proportion of THP-1/MCF-7 tumor hybrid cells in co-cultures supplemented with 40 µg/ml and 80 µg/ml CD36 antibody was 1.8% and 1.9%, respectively. In non-treated THP-1/MCF-7 co-cultures, the proportion of tumor hybrid cells (3.6%) was significantly higher compared to co-cultures treated with anti-CD36 antibody (P<0.001).

association with tumor progression. However, our understanding of the cellular processes leading to tumor cell fusion remains limited. In the present study, we explore how ionizing radiation affects the expression of CD36 and PS in THP-1 and MCF-7 cells and how it influences the fusion events between these cells. The spontaneous THP-1/MCF-7 tumor hybrid cell generation occurred at an average rate of 2.8%, but after irradiation, the fusion rate significantly increased to 6%. The proportion of MCF-7 cancer cells expressing PS increased proportionally with the radiation dose and the time elapsed after radiation exposure. Notably, the expression of CD36 in monocytes remained unaffected by radiation. Treatment of cells with anti-CD36 antibodies resulted in a significant reduction in THP-1/MCF-7 fusion rates.

## Cell fusion in cancer

The spontaneous fusion between cancer and stroma cells is an active process that requires the participation of both cell types and comprises several steps, including cell adhesion, membrane rearrangement, and post-fusion reorganization of the cells into a syncytium [39]. Phosphatidylserine plays a crucial role in cell-to-cell fusion, particularly in the initial fusion stage during cell adhesion. For instance, exposed PS contributes to the formation of multinucleated skeletal muscle and myoblasts independently of apoptotic changes or caspase activity in myocytes [40–42]. Phosphatidylserine is also involved in macrophage fusion and requires the CD36 receptor to recognize and mediate the capture of PS on other fusing cells [43, 44].

Cell fusion is a complicated multistep process that involves the following steps: programming target cells to make them competent for fusion, chemotaxis, intercellular recognition and attachment, and cell membrane fusion [45]. Our current study demonstrates that inhibiting the interaction between CD36 and PS by blocking the CD36 receptor resulted in a significant reduction in fusion events between THP-1 and MCF-7 cells. These findings provide compelling evidence that CD36 and PS signaling interplay is involved in the recognition and

attachment steps during the fusion process between monocytes and cancer cells. Accumulating evidence supports the notion that cell fusion contributes to tumor progression through the hybridization of cancer cells with bone marrow-derived cells, such as monocytes/macrophages, resulting in tumor hybrid cells with a significantly faster growth rate *in vivo* [20, 46] and enhanced abilities in colony formation [46], cell migration and invasion [47]. Consequently, cell fusion generates new metastatic cancer cell clones with tumor growth-promoting properties [21, 48–52]. Hence, understanding the cell fusion process as a contributing mechanism in cancer development is crucial for identifying novel targeted therapy concepts in cancer treatment.

## The clinical significance of leukocyte/cancer cell fusion

Even though cell fusion is a rare phenomenon, and the proportion of tumor hybrid cells may be small compared to the total number of cells in the tumor microenvironment, these tumor hybrid cells might constitute a crucial factor in cancer metastasis. Based on the assumption that one gram or one $cm^3$ of tumor mass contains approximately $1x10^9$ tumor cells [42, 53], a cell fusion rate of about 2% would potentially generate around twenty million hybrid cells in each gram or $cm^3$ of solid tumor. Moreover, the fusion events significantly escalate under pathological conditions such as inflammation [54–56]. Tumor-related inflammation is a crucial characteristic of cancer, and there is a well-established correlation between chronic inflammation and tumor development [57]. The cellular mechanisms contributing to the increase in cell fusion rates induced by inflammation or irradiation are currently unknown. In this study, we demonstrate that irradiation leads to a significant dose-dependent increase in the number of cancer cells expressing PS. At the same time, it does not affect the expression of CD36 in either THP-1 or MCF cells. Moreover, the irradiation of these cells results in a marked rise in THP-1/MCF-7 fusion events. These findings suggest that increased monocyte/cancer cell fusion during pathological processes, such as inflammation, radiation, or tissue damage, might be mediated by enhanced PS expression in cancer cells.

CD36 plays a significant role in the development and progression of various cancer types, including breast cancer [58], acute myeloid leukaemia [59], and gastric cancer [60], through several mechanisms, such as the activation of cancer stem cells, epithelial-to-mesenchymal transition, and enhancing chemoresistance [61, 62]. CD36-induced metastasis is known to be linked to the glucose and lipid metabolism of cancer cells and mediated through the GSK-3β/β-catenin signaling pathway [34, 63, 64]. Cancer cells expose PS in the outer cell membrane layer without displaying any sign of apoptosis [65–67]. The exposure of PS by tumor cells results in immune suppression and promotes tumor growth [68, 69]. However, the reason why cancer cells expose PS to the outer cell membrane leaflet without undergoing apoptosis remains unknown. Both PS and CD36 are suggested as potential therapeutic targets for cancer treatment [70–72]. For example, Bavituximab, a chimeric monoclonal antibody that targets PS, is believed to induce antitumor effects by activating T-cell-driven immune pathways against the tumor cells [73]. According to the cell fusion theory, preventing monocyte/cancer cell fusion with anti-PS or anti-CD36 antibodies could potentially impair the generation of new metastatic cancer cell clones and impede tumor progression and metastasis.

## Conclusion

Irradiation induces a dose-dependent increase in cancer cells expressing PS without affecting CD36 expression in THP-1 or MCF-7 cells. Fusion between MCF-7 and THP-1 cells increases following irradiation. Blocking CD36 and impairing PS-CD36 interaction significantly reduces fusion events between MCF-7 and THP-1 cells. To the best of our knowledge, this is the first

study to elucidate the role of PS-CD36 interaction in monocyte/cancer cell fusion, suggesting that the rising of fusion events might be mediated by increased expression of PS in cancer cells.

## Supporting information

**S1 File.**
(SAV)

**S1 Data.**
(XLSX)

## Acknowledgments

We want to thank Dimitrios Kalafatidis at the Clinic of Oncology at the University Hospital in Linköping for his help with the irradiation of cell cultures. We are also grateful to Dr Tony Forslund for providing THP-1 cells and Professor Olle Stål for providing MCF-7 cells.

## Author Contributions

**Conceptualization:** Ivan Shabo, Kristine Midtbö, Annelie Lindström.

**Formal analysis:** Ivan Shabo, Kristine Midtbö, Annelie Lindström.

**Investigation:** Ivan Shabo, Kristine Midtbö, Annelie Lindström.

**Methodology:** Ivan Shabo, Kristine Midtbö, Annelie Lindström.

**Project administration:** Ivan Shabo, Annelie Lindström.

**Software:** Ivan Shabo.

**Supervision:** Ivan Shabo, Annelie Lindström.

**Visualization:** Robert Bränström.

**Writing – original draft:** Ivan Shabo.

**Writing – review & editing:** Ivan Shabo, Robert Bränström, Annelie Lindström.

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
