## [Decision Letter · Decision Letter 0]

20 Mar 2024

PONE-D-23-37839Monocyte-cancer cell fusion is mediated by phosphatidylserine - CD36 receptor interaction and induced by ionizing radiationPLOS ONE

Dear Dr. Shabo,

Thank you for submitting your manuscript to PLOS ONE. After careful consideration, we feel that it has merit but does not fully meet PLOS ONE’s publication criteria as it currently stands. Therefore, we invite you to submit a revised version of the manuscript that addresses the points raised during the review process.

We look forward to receiving your revised manuscript.

Kind regards,

Ajay Kumar, PhD

Academic Editor

PLOS ONE

Journal Requirements:

- file:///home/nkw-ld22-090/Downloads/WJCOv11i3.pdf

In your revision ensure you cite all your sources (including your own works), and quote or rephrase any duplicated text outside the methods section. Further consideration is dependent on these concerns being addressed.

"The Research Council of Southeast Sweden Projekt ID: FORSS-229761"

4. In the online submission form you indicate that your data is not available for proprietary reasons and have provided a contact point for accessing this data. Please note that your current contact point is a co-author on this manuscript. According to our Data Policy, the contact point must not be an author on the manuscript and must be an institutional contact, ideally not an individual. Please revise your data statement to a non-author institutional point of contact, such as a data access or ethics committee, and send this to us via return email. Please also include contact information for the third party organization, and please include the full citation of where the data can be found.

Reviewers' comments:

Reviewer's Responses to Questions

**Comments to the Author**

1. Is the manuscript technically sound, and do the data support the conclusions?

Reviewer #1: Yes

Reviewer #2: Partly

Reviewer #3: Partly

2. Has the statistical analysis been performed appropriately and rigorously? 

Reviewer #1: Yes

Reviewer #2: Yes

Reviewer #3: Yes

3. Have the authors made all data underlying the findings in their manuscript fully available?

Reviewer #1: Yes

Reviewer #2: Yes

Reviewer #3: Yes

4. Is the manuscript presented in an intelligible fashion and written in standard English?

Reviewer #1: Yes

Reviewer #2: Yes

Reviewer #3: Yes

5. Review Comments to the Author

Reviewer #1: In the present investigation “Monocyte-cancer cell fusion is mediated by phosphatidylserine - CD36 receptor interaction and induced by ionizing radiation”, Shabo et al have reported ionizing radiation resulted in the fusion of monocyte and tumor cells through the interaction between phosphatidylserine and CD36. The authors have co-cultured the tumor cell line and monocyte cell line and were subjected to ionizing radiation in the presence and absence of anti-CD36 antibodies to evaluate the interaction between phosphatidylserine and CD36 in the fusion of tumor cells and monocyte.

There are many studies studies regarding the fusion of cancer cells among themselves or/and with macrophages. However, to make this study more interesting and impactful, following comments can substantiate their findings.

1. The authors have presented and have explained the findings suitably. However, the language of the manuscript requires more improvement.

2. The authors have used the flow cytometry to show the fusion of the cells. However, the fusion of the cells should also be confirmed through microscopy. Since macrophages and monocytes are the scavenging and are involved in the phagocytosis, hence, microscopic images will confirm that the tumor cells are being phagocytosed not the apoptotic cell bodies.

3. The author should also measure the oxidative stress in the irradiated tumor and monocyte cell lines and fused tumor cells and cancer cell lines.

4. CD-36 have a prominent role in the scavenging potential of macrophages and phagocytosis. Therefore, the authors should also show the scavenging potential of the monocyte cell line in the presence of anti-CD36 antibodies.

5. The authors should confirm their findings with monocytes isolated from the human blood which will cement their findings.

6. The authors should show viability of the tumor cells and monocytes before and after the irradiation.

Reviewer #2: This study represents a significant leap forward in unraveling the intricate mechanisms of leukocyte-cancer cell fusion. By highlighting the role of the CD36-phosphatidylserine interaction and its modulation by ionizing radiation, the research not only deepens our understanding of tumor progression but also opens avenues for targeted therapeutic strategies in combating metastatic cancer. However, there are some potential drawbacks and limitations that should be considered.

1. The investigation is exclusively conducted in vitro, posing a potential limitation in fully replicating the intricate dynamics of the in vivo microenvironment. Cellular interactions within the controlled laboratory conditions may diverge from those occurring within living organisms, possibly constraining the generalizability of the study's outcomes.

2. The utilization of specific cell lines (THP-1 and MCF-7) may not comprehensively represent the diverse array of cancer and immune cells present in authentic tumors. Different responses among various cancer types and subtypes may exist, and relying solely on a single breast cancer cell line might overlook this inherent variability.

3. While the study underscores the significance of CD36 and phosphatidylserine in the fusion of leukocytes with cancer cells, it is imperative to acknowledge that additional molecular factors and signaling pathways could also play contributory roles in this intricate process. A more all-encompassing analysis of these involved molecules could yield a more comprehensive understanding.

4. The primary focus of the study is on unraveling mechanisms at the cellular and molecular levels. Translating these discoveries into clinical applications necessitates further exploration to ascertain the clinical relevance and potential therapeutic implications of targeting CD36-phosphatidylserine interactions.

5. Despite the compelling nature of the in vitro findings, the absence of in vivo validation imposes constraints on directly applying the results to the complexities of biological systems. Subsequent studies incorporating animal models or clinical samples hold promise for providing a more precise representation of the physiological relevance of these findings.

6. The study predominantly delves into the impact of ionizing radiation on leukocyte-cancer cell fusion. While this aspect is pivotal, an exclusive focus on a single treatment modality may fall short in capturing the diverse conditions encountered by cancer patients undergoing various therapeutic interventions.

7. The study overlooks the heterogeneity prevalent in cancer patients, encompassing variations in tumor microenvironments, patient responses to treatment, and the dynamic nature of cancer progression. Acknowledging and understanding these diverse factors are crucial for extending the applicability of the findings to a broader clinical context.

Reviewer #3: Reviewer Comments:

The manuscript entitled " Monocyte-cancer cell fusion is mediated by phosphatidylserine - CD36 receptor interaction and induced by ionizing radiation” Current study highlights irradiation leads to a significant dose-dependent increase in the number of cancer cells expressing PS, while it does not affect the expression of CD36 in either THP-1 or MCF cells. Moreover, cell fusion generates new metastatic cancer cell/Tumor hybrid cells (THC) that exhibits the unique properties like abilities in colony formation, cell migration, invasion and over all acquired growth promoting properties. This is very an interesting study that can provide clue of targeted therapy for high grade tumor. However, I still have some queries and suggestion for the improvement of work qualities. Authors are advice to deal with the comments and adapt the manuscript accordingly and submit a revised manuscript. At this stage manuscript does not hold merit of acceptance. I would like to review the manuscript again.

Major concerns:

1. As a results of Monocyte-cancer cell fusion a tumor hybrid cells (THC) get formed which exhibits a unique features like abilities of colony formation, cell migration, and invasion. In this study author has simply analyzed all data by FACS ARIA III. Since data is very unique and important so it should be validated by multiple parameters like morphological assessment by Immunofluorescence/ high contrast laser scanning confocal microscopy (LSCM ) images and some physiological assessment like colony formation assay, cell migration and invasion assay is very essential for further validation.

2. In the study, a single MCF7 luminal breast cancer cells have been selected for the study, which is more prone to form hybrids tumor cells than any other cells. Is there any specific reason of choosing this cell? Though cell fusion is a rare phenomenon, and the proportion of hybrid cells may be small compared to the total number of cells. Therefore, it is advice to use other cell lines too for validation of the results along with normal cells.

Manor concerns:

1. It is better to use term tumor hybrid cells (THC) which is more specific and technical instead of cancer cell fusion in running text.

2. Almost 80 to 90% references given to the article is quite old, kindly updates with some recent articles related to fields. There are some references related to the field may be included i.e. PMID: 30339548, PMID: 32612123, PMID: 28870843.

6. PLOS authors have the option to publish the peer review history of their article (what does this mean?). If published, this will include your full peer review and any attached files.

Reviewer #1: **Yes: **Dr. VISHAL KUMAR GUPTA

Reviewer #2: No

Reviewer #3: **Yes: **Dr. Shyam Babu Prasad

---

## [Author Response · Author response to Decision Letter 0]

17 Apr 2024

As comments to reviewers, please see separate document "Response to reviviers".

---

## [Decision Letter · Decision Letter 1]

15 Jul 2024

PONE-D-23-37839R1Monocyte-cancer cell fusion is mediated by phosphatidylserine - CD36 receptor interaction and induced by ionizing radiationPLOS ONE

Dear Dr. Shabo,

Thank you for submitting your manuscript to PLOS ONE. After careful consideration, we feel that it has merit but does not fully meet PLOS ONE’s publication criteria as it currently stands. Therefore, we invite you to submit a revised version of the manuscript that addresses the points raised during the review process.

We look forward to receiving your revised manuscript.

Kind regards,

Ajay Kumar, PhD

Academic Editor

PLOS ONE

Journal Requirements:

Reviewers' comments:

Reviewer's Responses to Questions

**Comments to the Author**

1. If the authors have adequately addressed your comments raised in a previous round of review and you feel that this manuscript is now acceptable for publication, you may indicate that here to bypass the “Comments to the Author” section, enter your conflict of interest statement in the “Confidential to Editor” section, and submit your "Accept" recommendation.

Reviewer #1: (No Response)

Reviewer #3: All comments have been addressed

Reviewer #4: All comments have been addressed

2. Is the manuscript technically sound, and do the data support the conclusions?

Reviewer #1: Partly

Reviewer #3: Yes

Reviewer #4: Yes

3. Has the statistical analysis been performed appropriately and rigorously? 

Reviewer #1: Yes

Reviewer #3: Yes

Reviewer #4: No

4. Have the authors made all data underlying the findings in their manuscript fully available?

Reviewer #1: Yes

Reviewer #3: Yes

Reviewer #4: Yes

5. Is the manuscript presented in an intelligible fashion and written in standard English?

Reviewer #1: Yes

Reviewer #3: Yes

Reviewer #4: Yes

6. Review Comments to the Author

Reviewer #1: The study's findings are very interesting; however, they lack a mechanistic approach. The authors must have included the experiments and findings with this work during revision rather than keeping them for a future goal. The addition of these experiments would have increased the impact of the present finding and made this work more interesting to the readers.

Reviewer #3: The authors have addressed almost all the issues raised by the reviewer, but there is still some query:

Comments 1# Authors has added fluorescence cell images in a new figure (Fig 2), which is simple MCF7, THP1 cells, and THP-1 cells stained with CD36 images using cellTraceTM staining. Authors should provide the cells images showing CD36, PS expression, and images of ionizing radiation showing their impact on CD36 and PS expression in cancer cells. In each case merge images should be provided. Further if possible try to capture the tumor hybrid cells images.

Reviewer #4: Minor point

1. Statistical analysis shown in figure 5, 6 and 7 needs to be elaborated by including the error bar, significance or level of significance etc. Also include the number of replicates used in experiment.

2. Add the methodology of fluorescence microscope imaging in manuscript and include the magnification, scale bar etc. in figure 2.

7. PLOS authors have the option to publish the peer review history of their article (what does this mean?). If published, this will include your full peer review and any attached files.

Reviewer #1: No

Reviewer #3: **Yes: **Dr. Shyam Babu Prasad

Reviewer #4: No

---

## [Author Response · Author response to Decision Letter 1]

19 Jul 2024

Please see separate document "R2 Response to Reviewer"

---

## [Editor Report · Decision Letter 2]

12 Sep 2024

Monocyte-cancer cell fusion is mediated by phosphatidylserine - CD36 receptor interaction and induced by ionizing radiation

PONE-D-23-37839R2

Dear Dr. Shabo

We’re pleased to inform you that your manuscript has been judged scientifically suitable for publication and will be formally accepted for publication once it meets all outstanding technical requirements.

Kind regards,

Ajay Kumar, PhD

Academic Editor

PLOS ONE

Additional Editor Comments (optional):

The authors have addressed all the comments of the reviewers, so, it can be accepted for the publication.
---

## [Editor Report · Acceptance letter]

1 Nov 2024

PONE-D-23-37839R2 

PLOS ONE

Dear Dr. Shabo, 

I'm pleased to inform you that your manuscript has been deemed suitable for publication in PLOS ONE. Congratulations! Your manuscript is now being handed over to our production team.

Kind regards, 

on behalf of

Dr. Ajay Kumar 

Academic Editor

PLOS ONE